# Maternal Acylcarnitine Disruption as a Potential Predictor of Preterm Birth in Primigravida: A Preliminary Investigation

**DOI:** 10.3390/nu16050595

**Published:** 2024-02-22

**Authors:** Ying-Chieh Han, Katarina Laketic, Kylie K. Hornaday, Donna M. Slater, Chunlong Mu, Suzanne C. Tough, Jane Shearer

**Affiliations:** 1Department of Biomedical Engineering, Faculty of Engineering, University of Calgary, 2500 University Dr. NW, Calgary, AB T2N 1N4, Canada; 2Department of Physiology and Pharmacology, Cumming School of Medicine, University of Calgary, 2500 University Dr. NW, Calgary, AB T2N 1N4, Canada; kylie.hornaday@ucalgary.ca (K.K.H.); dmslater@ucalgary.ca (D.M.S.); 3Alberta Children’s Hospital Research Institute, Cumming School of Medicine, University of Calgary, 2500 University Dr. NW, Calgary, AB T2N 1N4, Canada; chunlong.mu1@ucalgary.ca (C.M.); stough@ucalgary.ca (S.C.T.); 4Department of Biochemistry and Molecular Biology, Cumming School of Medicine, University of Calgary, 3330 Hospital Drive NW, Calgary, AB T2N 4N1, Canada; 5Faculty of Kinesiology, University of Calgary, 2500 University Dr. NW, Calgary, AB T2N 1N4, Canada; 6Department of Pediatrics and Community Health Sciences, Cumming School of Medicine, University of Calgary, 2500 University Dr. NW, Calgary, AB T2N 1N4, Canada

**Keywords:** pregnancy, metabolism, primigravida, screening, metabolomics, mass spectrometry, predictive modelling, fatty acid

## Abstract

Preterm birth, defined as any birth before 37 weeks of completed gestation, poses adverse health risks to both mothers and infants. Despite preterm birth being associated with several risk factors, its relationship to maternal metabolism remains unclear, especially in first-time mothers. Aims of the present study were to identify maternal metabolic disruptions associated with preterm birth and to evaluate their predictive potentials. Blood was collected, and the serum harvested from the mothers of 24 preterm and 42 term births at 28–32 weeks gestation (onset of the 3rd trimester). Serum samples were assayed by untargeted metabolomic analyses via liquid chromatography/mass spectrometry (QTOF-LC/MS). Metabolites were annotated by inputting the observed mass-to-charge ratio into the Human Metabolome Database (HMDB). Analysis of 181 identified metabolites by PLS-DA modeling using SIMCA (v17) showed reasonable separation between the two groups (CV-ANOVA, *p* = 0.02). Further statistical analysis revealed lower serum levels of various acyl carnitines and amino acid metabolites in preterm mothers. Butenylcarnitine (C4:1), a short-chain acylcarnitine, was found to be the most predictive of preterm birth (AUROC = 0.73, [CI] 0.60–0.86). These observations, in conjuncture with past literature, reveal disruptions in fatty acid oxidation and energy metabolism in preterm primigravida. While these findings require validation, they reflect altered metabolic pathways that may be predictive of preterm delivery in primigravida.

## 1. Introduction

Impacting approximately 10% of births globally, it is estimated that over 15 million babies are born prematurely each year [1,2]. Premature birth is the primary cause of mortality in children, responsible for 18% of all fatalities in those under the age of five years. It also results in adverse health risks to both mothers and infants and is the leading cause of lifelong physical and neurodevelopmental disabilities [3].

Studies have identified several risk factors related to preterm birth, ranging from psychological, physiological to environmental aspects. Current preterm risk assessment utilizes multiple diagnostic approaches based on past-pregnancy history, cervical length measurements and biomarkers. While these tools have demonstrated predictive potential and have improved outcomes, they can be limited by factors such as first-time (primigravid) pregnancy or the sensitivity of measurements [4]. As such, there are ongoing efforts to enhance predictive accuracy of preterm birth as well as to refine interventions for better maternal and neonatal outcomes.

Advancements in bioanalytical technologies have enabled the simultaneous detection of trace compounds in biological matrices and sparked interest in their application to metabolomic analyses related to gestation. These include the investigation of gestational weight gain [5], birth weight [6], exercise [7] and preeclampsia [8]. To date, metabolomic predictors of preterm birth are highly heterogeneous, largely owing to differences in study design, population, outcome definitions (e.g., preterm birth), and biospecimen type (serum/plasma, maternal cord sample, placenta). Despite these limitations, metabolites associated with preterm birth have been found, including maternal lipids [9], volatile organic acids [10], and placental tryptophan metabolites indicative of inflammation [11]. In the present study, we sought to retrospectively examine the relationship between maternal serum metabolomics and preterm delivery in primigravid women. This group is unique, as they frequently do not show any risk factors or warning symptoms of preterm birth and reproducible screening tools are currently not available in clinical practice. Employing a well-established birth cohort, we tested the hypothesis that untargeted metabolomics of maternal serum early in the third trimester may be predictive in distinguishing eventual preterm and term births. 

## 2. Materials and Methods

### 2.1. Participants and Sample Collection

The All Our Families pregnancy cohort (formally All Our Babies) is an ongoing longitudinal study monitoring representative mother–baby pairs in Calgary, Alberta [12]. This study was approved by the Child Health Research Office and the Conjoint Health Research Ethics Board at the University of Calgary, and written informed consent was obtained from all participants. Eligibility was based on the individuals being adults (≥18 years of age), being at less than 25 weeks of gestation at the time of recruitment, receiving prenatal care in Alberta, Canada and being willing to complete written questionnaires in English. Maternal blood samples were collected from non-fasted primigravid participants from the antecubital vein at 28–32 weeks gestation between 2008 and 2011. Gestational age was calculated based on the participants’ last reported menstrual periods. Serum was centrifuged and stored at −80 °C until further analysis. Human serum metabolome has been found to remain largely stable at −80 °C within 2 freeze/thaw cycles, which was within the scope of the present study [13]. Primigravid status, namely first-time pregnancy, was determined based on the self-reported questionnaire and/or medical history. Preterm birth status and other clinical information were obtained from the database for participant classification. The statistical power of this study was not calculated a priori due to the exploratory nature of the study and analysis conducted. Within the cohort, preterm primigravida were age, pre-pregnancy BMI (kg/m^2^) and gestational weight-gain matched with those with primigravida term births in a 1:2 ratio for analysis. As mental health is a risk factor for preterm birth [14,15], scores of the Spielberger State-Trait Anxiety Inventory Score (STAI) and Edinburgh Perinatal Depression Scale (EPDS), as well as probable depression and anxiety, were also reported. 

### 2.2. Sample Preparation and Measurement

Investigators were blinded to sample identity until all spectra were collected and processed. In 2020, serum samples were assessed as previously described [16]. Briefly, frozen serum samples were thawed on ice from −80 °C storage, and 50 µL of each sample was transferred into a 1.7 mL polypropylene vial prior to protein precipitation using 200 µL of 100% methanol. The mixture was vortexed on a benchtop unit (Fisherbrand, Fisher Scientific, Pittsburgh, PA, USA) set at the maximum level for 1 min and centrifuged (Eppendorf Canada, Mississauga, ON, Canada. Model 5424R) at 14,000 rpm (18,407 g) for 10 min at 4 °C to condense the precipitate. The supernatants were collected into a new set of tubes using transfer pipettes followed by solvent evaporation to complete dryness using a vacuum evaporator at 40 °C. Dried samples were reconstituted in 100 µL of 50:50 methanol/water and vortex mixed until all residues visibly dissolved. The solutions were then centrifuged through a 200-micron filter at 14,000 rpm for 10 min at 4 °C to remove any potential debris. 

A displacement pipette was then used to transfer 90 µL of collected samples into glass LC/MS vials with caps before storing them at −80 °C until the day of measurement. Blank serum samples collected from healthy, non-pregnant volunteers were assayed following the same procedure to monitor and evaluate the subsequent instrument performance. Samples were analyzed via untargeted metabolomics using positive mode liquid chromatography mass spectrometry (LC/MS, QTOF 6545i, Agilent, Santa Clara, CA, USA). An autosampler compartment was set at 4 °C to maintain the sample vials at a refrigerated temperature. Reverse phase liquid chromatography was performed through an Acquity HSS (2.1 × 150 mm, Waters, Milford, MA, USA) at a temperature of 40 °C for better partitioning under a mobile phase system of (A) 0.1% formic acid in water and (B) 0.1% formic acid in acetonitrile. A gradient elution at a constant flow rate of 0.400 mL/minute was applied to separate compounds based on their chemical affinities using the following time scheme of mobile phase (A), and (B) elution was initiated with 5% B for 1.5 min, then a linear gradient of B from 5% to 100% for 14 min, followed by 100% B for 3 min. Lastly, the gradient returned to the 5% B starting condition at the 17 min mark and equilibrated for 2 min to conclude the run. The overall run time was 19 min gradient, plus 1.5 min of instrument lag time, for a total of 20.5 min. Blank samples were injected prior to and in between sample injections to assess instrument stability and reproducibility.

### 2.3. Metabolite Identification and Statistical Analysis

LC/MS spectra were collected via MassHunter software version 10.0 (Agilent, Santa Clara, CA, USA). Each datafile was converted to the mzML file format using MSConvert from the ProteoWizard version 3.0.6150 software package. Collected spectra were uploaded and processed through XCMS version 3.7.1 for peak labelling and intensity calculations [17]. Metabolites were identified by inputting the observed mass-to-charge (*m*/*z*) ratio between 50 and 1200 *m*/*z* into the Human Metabolome Database (HMDB) and determining the most probable compound candidate with a tolerance threshold of 30 ppm. A total of 3408 peaks were obtained from XCMS following the exclusion of signals below detection limits and signals with less than 50% presence across all samples. Compounds with a CV > 30% were excluded from further analyses [18]. In total, 181 endogenous, known metabolites were identified. Drugs and other molecules belonging to the human exposome were not taken into consideration during compound annotation. For metabolites, data were centered and log-transformed to improve symmetry and approximate normality. To gain an appreciation for global differences in metabolite patterns between preterm and term groups, data were modelled using partial least squares-discriminant analysis (PLS-DA) for multivariate modeling using SIMCA version 17.0 (UMetrics, Umea, Sweden). Metabolite contribution in model performance was determined based on the variable importance in projection (VIP) scores for the most dominant components [19]. All metabolites with a VIP > 1 were subsequently analyzed by a Student’s *t*-test to identify statistical significance (*p* < 0.05) between the preterm and term mothers. To control the family-wise false positive rate, a Benjamini-Hochberg (FDR) correction was applied to the 181 identified metabolites [20]. An FDR threshold of q < 0.15 was selected for a more liberal evaluation in this explorative pilot study. Predictive potentials of individual metabolites in preterm birth were evaluated using an area under the receiver-operating characteristic (AUROC) curve, while the confidence interval (CI) and the cut-off sensitivity and specificity were generated by the Wilson–Brown method [21] (GraphPad Prism 9.0, Boston, MA, USA).

## 3. Results

### 3.1. Participant Characteristics

Participant characteristics within the preterm and term groups are shown in Table 1. Within the analyzed cohort, the only significant differences between preterm and term groups were gestational age and birthweight (*p* < 0.05); there were no differences in maternal age, pre-pregnancy BMI (kg/m^2^) or gestational weight gain between groups. Likewise, mental health assessments also did not report differences in the prevalence of anxiety or depression between preterm and term primigravida. There was also an upward trend, though not statistically significant (*p* = 0.073), for incidence of caesarean sections (excluding voluntary choice) among preterm primigravida (41.7%), indicating greater need for physician intervention within this group. Specific presentations and reasons for intervention (e.g., induction, c-section, etc.) from medical records are listed. Due to the limited sample size, the preterm group in our analysis included any participant who delivered before 37 weeks of gestation and did not further differentiate between very early preterm (<32 weeks) and mid–late preterm birth (32 to 37 weeks), as defined by the World Health Organization [22]. 

### 3.2. Metabolite Profiling

The majority of peaks were eluted before the 12-min mark at the end of the linear gradient on the spectra (Figure 1), proving effective trap/release of the current setup. The observed number of peaks differ across samples. Among these, 3408 peaks were consistently present in over half of all samples and were subjected to subsequent analysis. Analysis of 181 identified metabolites by PLS-DA using a three-component model showed reasonable separation between the two groups (CV-ANOVA, *p* = 0.02; Figure 2A, Appendix A). From this model, there were 52 identified metabolites with a VIP > 1. Within this list, there were 14 metabolites that were significantly different (VIP > 1 and q < 0.15) between preterm and term mothers. Most of the identified metabolites (Figure 2B) fell into two broad categories: amino acids and amino acid relatives (e.g., alanine, glutaminyl-lysine, acetyl-L-arginine), and short and medium chain acylcarnitines.

A fold-change of metabolites is s represented by volcano plot (Figure 3A). Among significant metabolites, acylcarnitines were the predominant class making up over half (9 out of 14) of the observed compounds. In addition, the relative comparison after z-score normalization (Figure 3B) showed consistent reductions among the identified acylcarnitine species.

### 3.3. Predictive Modelling

The most significant metabolite predictive of preterm birth, which was also the compound with the largest fold differential at 2.09 was butenylcarnitine (C4:1), a butenoic acid ester of carnitine (Figure 4A,B). A receiver-operating characteristic curve for butenylcarnitine was used to evaluate the predictive ability of this metabolite. Results show that this metabolite was predictive of pre-term birth (AUC = 0.73, CI: 0.60–0.86, *p* < 0.05), with reasonable specificity (0.64) and sensitivity (0.83). Additionally, all nine of the significant acylcarnitines showed predictive potentials (*p* < 0.05) from their ROC results, all with an AUC between 0.67–0.74, Table 2.

## 4. Discussion

Preterm birth is a critical public health issue of global magnitude [23]. Therefore, it is crucial to develop tools to both predict and understand the pathophysiology of preterm birth. The present study retrospectively analyzed maternal serum collected in the third trimester in primigravid pregnancies in an established birth cohort to evaluate metabolomic predictors of preterm birth. Key findings show metabolic disruptions in primigravid women who experienced preterm delivery, including lower levels of amino acids and their derivatives, as well as specific acylcarnitines.

Acylcarnitines were among the compounds with the largest differentials and accounted for half of the observed significantly different metabolites with preterm birth. Acylcarnitines transport acyl groups from the cytosol into the mitochondrial matrix to produce energy via beta oxidation to sustain cellular energetics. To date, over 1200 acylcarnitine species have been identified, many with unknown roles and functions. Of the acylcarnitine species, the short-chain (C2-C5) are the most abundant, representing approximately 80% of all acylcarnitines [24]. During pregnancy, carnitine/acylcarnitine is essential for fetal growth, yet fetal carnitine synthesis increases proportionally with gestational age, meaning the need for carnitine/acylcarnitines must be supplied maternally through the placenta over the course of gestation [25]. As a result, there are large variations in maternal acylcarnitines as pregnancy proceeds, with a generalized increase in medium chain (C6-C12), but a marked decrease in both short (C2-C5) and long (C13-C20) chain acylcarnitines [26]. Reasons for this varied response are unknown and may involve the specific need or preference of the placenta/fetus for certain acyl lengths/species over others throughout gestation. Results of the present study show reductions in nine short- and medium-chain acyl carnitines in preterm mothers. Several specific acylcarnitines were of interest due to previous reports of their disruption in pregnancy and pregnancy-related conditions [27,28].

Our results show that the levels of serum dodec-enedioylcarnitine, the carboxylated derivative of dodecenoylcarnitine C12:1, were lower in preterm compared to term mothers. Of interest, the ratio of dodecanoylcarnitine/dodecenoylcarnitine (C12/C12:1) has been previously shown to associate with the occurrence of placental abruption and a maternal history of abnormal, early pregnancy vaginal bleeding, fitting with the present finding [27]. Likewise, acylcarnitines of C8, C10 and C12 have been shown to be disrupted in the first trimester serum of women who developed pre-eclampsia and delivered prior to 34 weeks gestation [28]. Butenylcarnitine (C4) is also known to be elevated in individuals with prepregnant obesity [29] as well as women who developed impaired glucose tolerance or type 2 diabetes following gestational diabetes mellitus [30]. Comparing the present study with similar research from the literature, previous work by de la Barca and colleagues reported elevations of short-chain acylcarnitines in the cord blood of intrauterine growth restricted newborns whose severity has been associated with how prematurely they were delivered [31]. In this study, increases in both short-chain and long-chain acylcarnitines were also observed in the placenta, and it was concluded that there were potential alterations in both lipid remodeling and fatty acid oxidation. This finding is further supported by the research of Elshenawy et al., who investigated the maternal side of the placenta and reported elevations in 17 of 41 acylcarnitines in spontaneous preterm birth. The majority of acylcarnitines were short-chain and included an ester of C4 [32]. This study linked accumulated acylcarnitines to reduced fatty acid oxidation capacity in the placenta, suggesting impaired mitochondrial function as a potential mechanism. Of interest, our findings observed a reverse trend for these metabolites in the maternal serum, suggesting that these compounds may accumulate differently in the maternal circulation versus the placenta in preterm birth. Such discrepancies give rise to new hypotheses and warrant additional investigation to better understand the mechanisms behind fetal–maternal interactions. Future studies are required to decipher the mechanisms behind these alterations and whether preventative measures or dietary supplementation would be of value. 

In addition to acylcarnitines, several amino acids and their derivatives were also distinct in preterm birth that may be indicative of greater uptake by the fetus or early placental dysfunction. Our finding of lower acetyl-L-arginine in preterm mothers is notable, as L-arginine and the nitric oxide system play an essential role in vascular compliance and placental function, upon which pregnancy outcomes may be dependent [33]. Dietary arginine assessed by 24 h food recall has been negatively correlated with preterm birth risk, with those in the highest consumption quartile having 0.79 times the risk of preterm birth prior to 37 weeks compared to the lowest quartile [34]. L-arginine supplementation also has favorable impacts on birth outcomes, reducing the risk of intrauterine growth retardation and preterm birth [35,36]. As a result, supplementation has been proposed as a recommendation for women at risk of pre-term birth, poor pregnancy outcomes, pre-eclampsia, and other hypertensive disorders of pregnancy [37]. However, in the case of primigravidas, where information regarding preterm birth risk may be limited, defining and testing normative acetyl-L-arginine ranges may be of use [38]. 

A main strength of this study is that it evaluated untargeted metabolomics of the maternal metabolome in the third trimester, the final stage of gestation prior to delivery, thus providing a final opportunity for possible intervention. It is also unique in that it examined primigravid versus multi-gravid pregnancies, a distinction rarely examined in the current literature. Limitations include a relatively small sample size, the non-fasted nature of the samples collected, and a lack of dietary controls. Different types of preterm delivery (e.g., spontaneous versus medically indicated preterm birth) were also not accounted for and could be clinically relevant. Next steps are to validate the findings in a separate birth cohort with a larger number of participants.

## 5. Conclusions

Overall, this study identified metabolic disruptions in primigravid women who experienced preterm delivery. Results show preterm mothers had altered amino acid metabolism and depleted levels of acylcarnitines, suggesting alterations in fatty acid oxidation. While these results require validation, these indicators may reflect altered metabolic pathways that may contribute to preterm delivery in primigravida.

## Figures and Tables

**Figure 1 nutrients-16-00595-f001:**
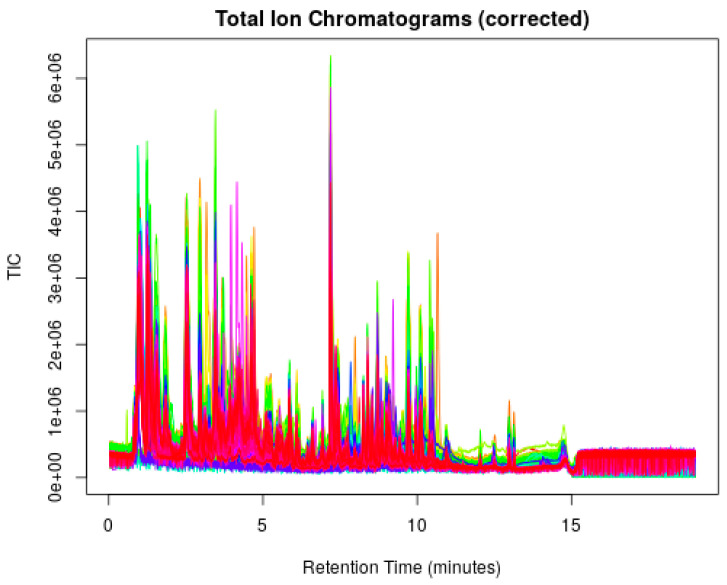
Total ion chromatogram (TIC) overlaying all samples across the spectra. Each color designates a sample injection. The spectra have been corrected for retention time deviations for peak alignments.

**Figure 2 nutrients-16-00595-f002:**
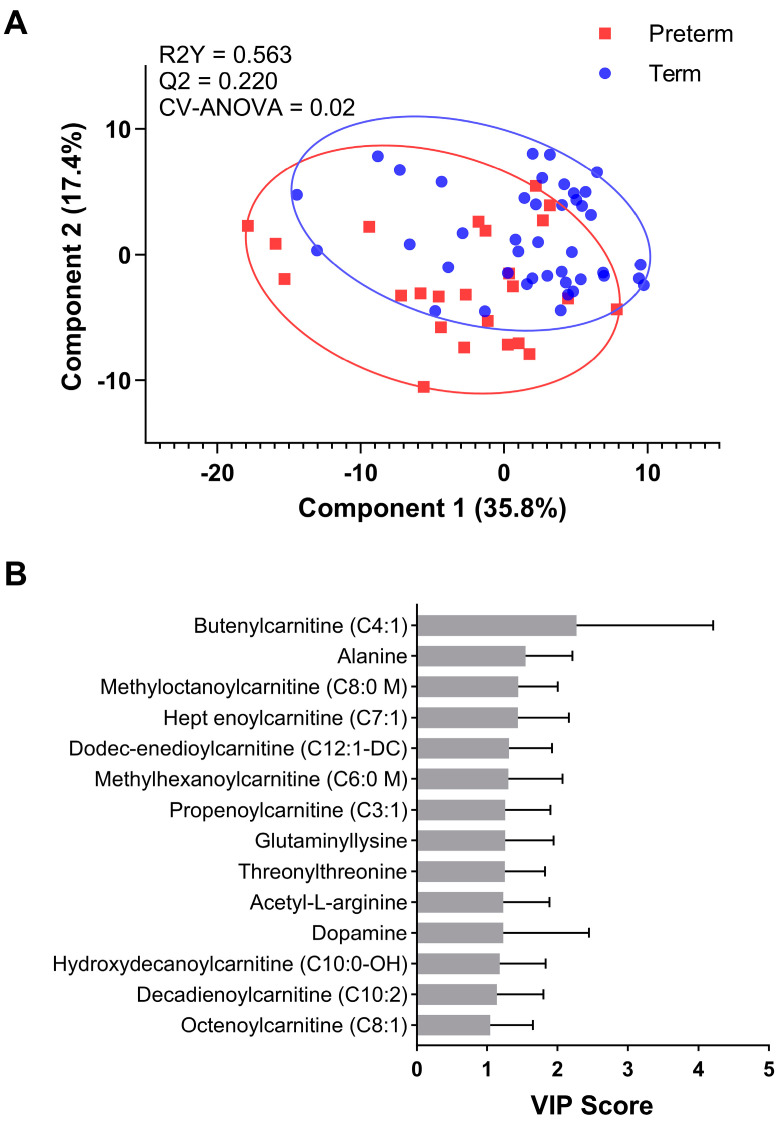
(**A**). Partial Least Squares-Discriminant Analysis (PLS-DA) plot of preterm (*n* = 24) and term (*n* = 42) among primigravida. CV−ANOVA = 0.02. (**B**). Variable Importance in Projection (VIP) scores > 1 modelled from the term versus preterm PLS-DA plot for metabolites with q < 0.15 to evaluate for further statistical significance.

**Figure 3 nutrients-16-00595-f003:**
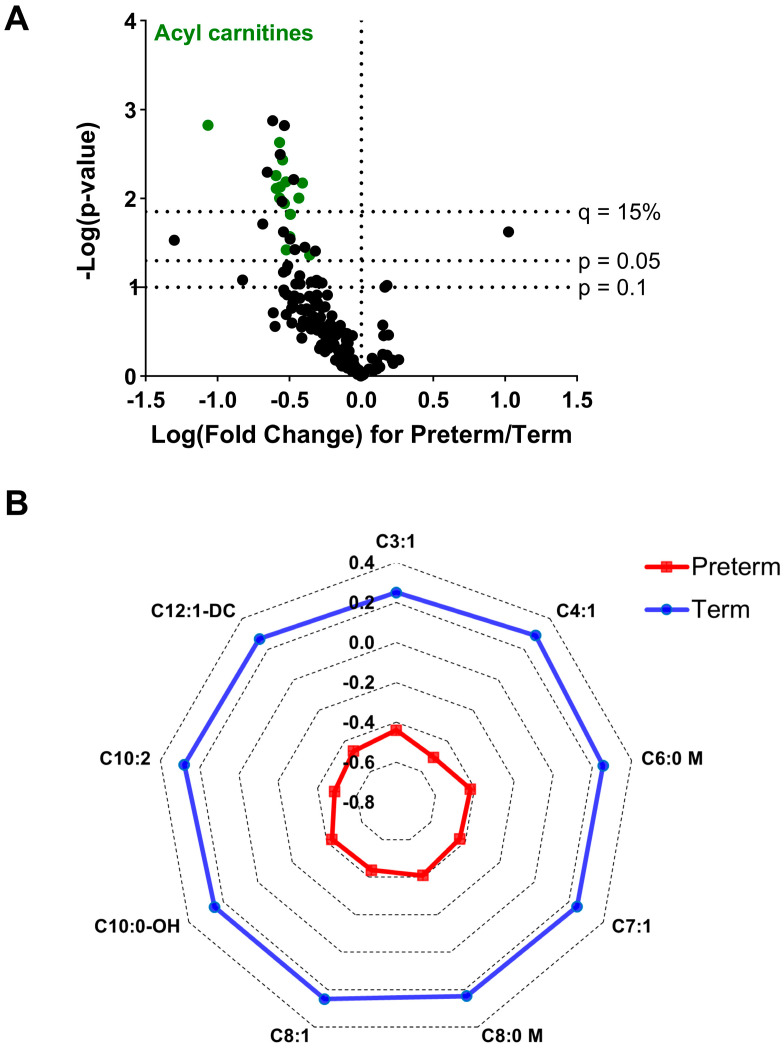
(**A**). Volcano plot for metabolic differences comparing preterm and term groups with each dot represent an annotated compound. Student’s *t* test with a two-tailed distribution was used to determine the *p*-value for each metabolite. Fold-change was calculated based on the mean peak intensity of preterm birth with respect to term birth. A threshold of 15% was chosen for false discovery assessment using the Benjamini and Hochberg method. (**B**). Radar plot showing z-scores of significant (q < 0.15) acylcarnitines (observed intensity−mean intensity)/standard deviation). Acyl carnitines are represented by their respective nomenclature.

**Figure 4 nutrients-16-00595-f004:**
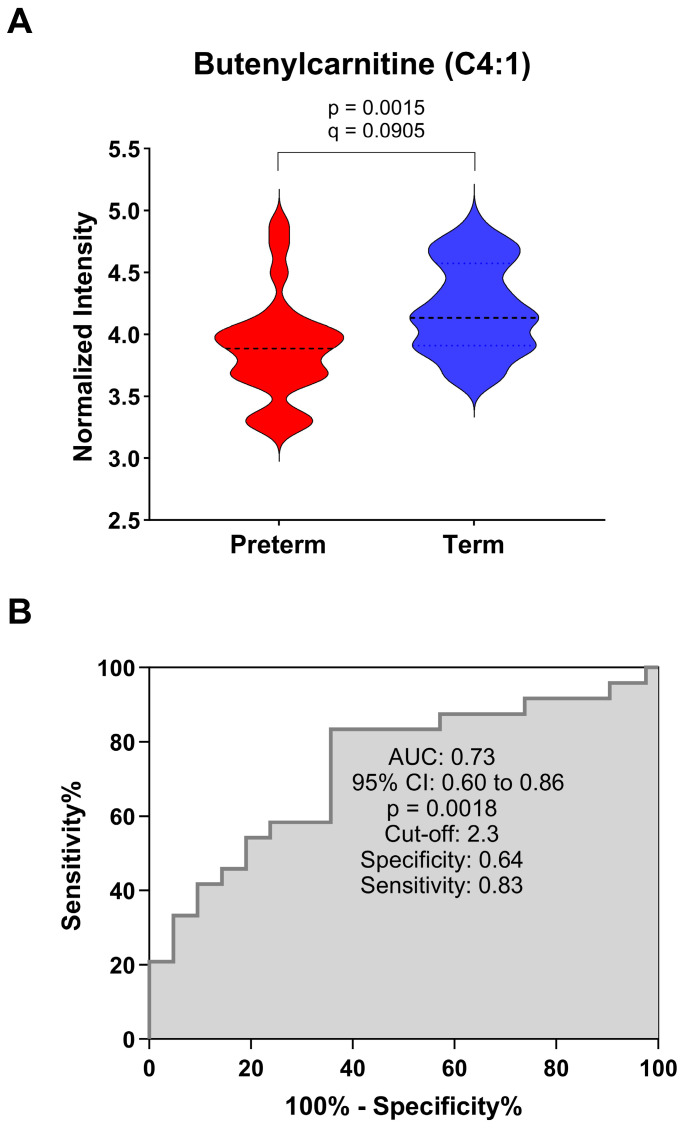
(**A**). Violin Plot indicating the distribution for butenylcarnitine between the preterm and the term groups. Significance is indicated. (**B**). Receiver-Operating Characteristic (ROC) curve for butenylcarnitine (C4:1) assessing the predictive potential for preterm delivery. Sensitivity and specificity were calculated using the normalized peak intensity for preterm (*n* = 24) and term (*n* = 42).

**Table 1 nutrients-16-00595-t001:** Maternal, Infant and Delivery Characteristics.

	Preterm (*n* = 24)	Term (*n* = 42)	*p*-Value
**Maternal**			
Pre-pregnancy BMI (kg/m^2^)	24.0 ± 6.2	23.9 ± 5.1	0.993
Maternal Age (y)	30.8 ± 4.2	29.6 ± 4.4	0.310
Gestation Weight Gain (kg)	12.4 ± 5.6	13.1 ± 6.0	0.311
Anxiety score ^1^	31.1 ± 6.7	30.8 ± 8.0	0.862
Depression score ^2^	5.7 ± 3.8	5.6 ± 4.4	0.927
Hypertensive ^3^ (*n*)(%)	3 (12.5)	2 (4.8)	0.319
**Infant**			
Gestation (wk)	34.3 ± 2.1	38.7 ± 1.1	<0.001
Fetal Birthweight (g)	2383 ± 612	3137 ± 342	<0.001
Child Sex, female (%)	10 (41.7)	22 (52.4)	0.461
**Delivery, *n* (%)**			
*C*-Section delivery	10 (41.7)	8 (18.2)	0.066
Spontaneous labor	10 (41.7)	25 (59.5)	0.220
Medically indicated (preterm)	12 (50)	-	-
Induction (term)	-	14 (33.3)	-
Heart rate abnormality ^4^	5 (20.8)	9 (21.4)	-
Premature placental separation	1 (4.2)	0 (0)	-
Breech position	3 (12.5)	1 (2.4)	-
Uterine prolapse	1 (4.2)	0 (0)	-
Premature membrane rupture	1 (4.2)	0 (0)	-
Restricted fetal growth	1 (4.2)	0 (0)	-
Uterine inertia	0 (0)	2 (4.8)	-
Incomplete fetal head rotation	0 (0)	2 (4.8)	-

Values represent mean ± standard deviation or categorical values (%). *p*-values were determined using an unpaired *t*-test with a two-tailed distribution. Significance set at *p* < 0.05. ^1^ Spielberger State-Trait Anxiety Inventory Score (STAI) was used to assess anxiety; ^2^ Edinburgh Perinatal Depression Scale (EPDS). ^3^ Gestational hypertension (>140/90 mmHg) after 20 weeks of pregnancy (with and without pre-eclampsia); ^4^ Clinical judgement, undefined.

**Table 2 nutrients-16-00595-t002:** Area Under the Receiver-Operating Characteristic for Significant Acylcarnitines.

Acylcarnitine	AUC	*p*-Value	95% CI
Decadienoylcarnitine (C10:2)	0.74	0.0023	0.61 to 0.86
Butenylcarnitine (C4:1)	0.73	0.0018	0.60 to 0.86
Propenoylcarnitine (C3:1)	0.70	0.0080	0.56 to 0.83
Dodec-enedioylcarnitine (C12:1-DC)	0.70	0.0083	0.56 to 0.84
Hydroxydecanoylcarnitine (C10:0-OH)	0.70	0.0066	0.57 to 0.84
Methylhexanoylcarnitine (C6:0 M)	0.70	0.0109	0.56 to 0.84
Hept enoylcarnitine (C7:1)	0.69	0.0101	0.55 to 0.83
Methyloctanoylcarnitine (C8:0 M)	0.69	0.0113	0.55 to 0.83
Octenoylcarnitine (C8:1)	0.67	0.0204	0.53 to 0.81

## Data Availability

All raw data are available upon request. The data are being uploaded to the NIH Common Fund’s National Metabolomics Data Repository (NMDR, https://www.metabolomicsworkbench.org/data/index.php (accessed on 25 March 2022). SIMCA version 17.0 and GraphPad PRISM 9.0 are purchased via licenses. XCMS online is accessed through a registered user account at https://xcmsonline.scripps.edu/ (accessed on 17 March 2022). The Human Metabolome Database (HMDB) and MetaboAnalyst 5.0 are freely accessible at https://hmdb.ca/and https://www.metaboanalyst.ca/ (accessed on 25 March 2022).

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
