# Peer review of "Maternal Acylcarnitine Disruption as a Potential Predictor of Preterm Birth in Primigravida: A Preliminary Investigation"

_nutrients, 2024, doi:10.3390/nu16050595_

Round 1

Reviewer 1 Report

Comments and Suggestions for Authors

This manuscript describes a study that conducted untargeted metabolomics on serum from primigravida women who had preterm births (n=24) compared to women who had term births (n=44), who are part of the All Our Families pregnancy cohort. Blood samples were collected from the women at 28-32weeks gestation and analyzed retrospectively (after the women delivered). The study found differences in acylcarnitines and amino acids between women who delivered prematurely compared to those that delivered at term. Overall, the manuscript is well-written and addresses an interesting research question of whether there are circulating metabolites that distinguish people who have a preterm birth compared to those that have a full-term birth. Below are specific comments/suggestions.

·      Blood samples. Please include if the blood samples were collected in the fasting state and whether the people were told to not take their prenatal supplement before the collection. It is stated in the limitations that the samples were non-fasted but please include this in the methods section.

·      Were statistical adjustments for multiple comparisons made?

·      Given the small sample size please include some commentary on the statistical power of the study. It is mentioned as a limitation but some discussion on this is required. This is very important given the conclusions and interpretation of the study findings.

Author Response

REVIEW 1

  1. Blood samples. Please include if the blood samples were collected in the fasting state and whether the people were told to not take their prenatal supplement before the collection. It is stated in the limitations that the samples were non-fasted but please include this in the methods section.

RESPONSE:  Thank you for your careful review.  These details have not been added to the manuscript. The addition is as follows:

Line 78-79. Methods. ‘Maternal blood samples were collected from non-fasted primigravid participants from the antecubital vein at 28-32 weeks gestation between 2008 and 2011’.

  1. Were statistical adjustments for multiple comparisons made?

RESPONSE: To account for multiple comparison issues in our testing, we calculated the false discovery rate (FDR) with a threshold of 0.15, which is a more conservative approach than standard biomarker profiling studies (Lewitt et al., 2013; Wang et al., 2021), to address statistical error in our results. This is described as follows:

Lines 145-148: ‘To control the family-wise false positive rate, a Benjamini-Hochberg (FDR) correction was applied to the 181 identified metabolites [21]. An FDR threshold of q < 0.15 was selected for a more liberal evaluation in this explorative pilot study’.

References

Lewitt, P. A., Li, J., Lu, M., Beach, T. G., Adler, C. H., & Guo, L. (2013). 3-hydroxykynurenine and other Parkinson’s disease biomarkers discovered by metabolomic analysis. Movement Disorders, 28(12), 1653–1660. https://doi.org/10.1002/MDS.25555

Wang, Y., Jacobs, E. J., Carter, B. D., Gapstur, S. M., & Stevens, V. L. (2021). Plasma Metabolomic Profiles and Risk of Advanced and Fatal Prostate Cancer. European Urology Oncology, 4(1), 56–65. https://doi.org/10.1016/J.EUO.2019.07.005

  1. Given the small sample size please include some commentary on the statistical power of the study. It is mentioned as a limitation but some discussion on this is required. This is very important given the conclusions and interpretation of the study findings.

RESPONSE: Statistical power of this study was not calculated a priori due to the exploratory nature of the study. Sample size calculations take specific variables into consideration and then determine an ideal number of participants. In the present study, over 7100 metabolites signals were detected. Given this, choosing metabolites on which to base statistical power calculations on becomes problematic. Having stated this, we also recognize the preliminary nature of this work and the inherent limitations based on sample size. For this reason, we have altered the title of the study to read: Maternal Acylcarnitine Disruption as a Potential Predictor of Preterm Birth in Primigravida: A Preliminary Investigation. We have also expanded our limitations section as follows:

Line 380-383. ‘Statistical power of this study was not calculated a priori due to the exploratory nature of the study and analysis conducted. While a greater sample size would have been ideal, next steps are to validate the findings in separate cohort with a larger number of participants’.

Reviewer 2 Report

Comments and Suggestions for Authors

1.      Line 21:  This should read “blood was collected” and the serum harvested…

2.     Line 21:  Was the blood obtained from the infants or the mother or both?  This is unclear here and in Line 27 you state “preterm mothers”.  Please be specific. Suggested wording; “blood was collected, and the serum harvested from the mothers of 24….”

3.     Line 70:  Please state how you determined gestational age.

4.     Lines 73-74:  Please state how long serum was frozen prior to analysis.  If you have/are aware of any data on the stability for known metabolites/fatty. acids when serum is stored at -80 C for this duration, please cite here.

5.     Lines 87 and 88:  Although I am sure this data are in your previous publications (13, 14) but for ease of the reader, please state what speed and duration centrifugation was performed here.

6.     Lines 110-111:  Is there a reference citation for the VIP scoring process?

7.     Line 132:  Should this read < 32 weeks?

8.     Line 152:  I don’t understand this “Error…” statement. Also, see Lines 165-166.

9.     Line 216:  Small “t” here.

10.  Line 22:  Should “AUROC” be spelled out here as Tables should stand independent of text?

11.  Discussion section:  Can the author(s) add a paragraph commenting/speculating, based on their and the published data cited, if certain supplement strategies (e.g., carnitine; AAs consumption during pregnancy) might be/could be researched and of possible benefit in preventing/minimizing preterm birth.  How might the clinician think about theoretical/possible application of your data in the care of their pregnant patients?

Author Response

REVIEW 2

Thank you for your time and effort to improve this manuscript. Your comments have added both detail and clarity. 

  1. Line 21:  This should read “blood was collected” and the serum harvested…

RESPONSE: Thank you for the feedback. The statement has been corrected in the manuscript as follows:

Line 22-23. ‘Blood was collected, and the serum harvested from the mothers of 24 preterm and 44 term births at 28-32 weeks gestation (onset of the 3rd trimester).’

  1. Line 21:  Was the blood obtained from the infants or the mother or both?  This is unclear here and in Line 27 you state “preterm mothers”.  Please be specific. Suggested wording; “blood was collected, and the serum harvested from the mothers of 24….”

RESPONSE: This has been revised together with Comment #1.

  1. Line 70:  Please state how you determined gestational age.

RESPONSE: Thank you for the feedback. The explanation has been added in the manuscript:

Line 79-80. ‘Gestational age was calculated based on the participants last reported menstrual period’.

  1. Lines 73-74:  Please state how long serum was frozen prior to analysis.  If you have/are aware of any data on the stability for known metabolites/fatty. acids when serum is stored at -80 C for this duration, please cite here.

RESPONSE: Thank you for the constructive input regarding sample stability. Serum Samples were collected from 2008 – 2011 and stored frozen at -80C until they were assayed in 2020 for the current study. All Our Families (AOF) was a tightly controlled longitudinal study and sample collection/storage/integrity is a a top priority. For this reason, the study maintains individual aliquots for use. Quality checks are also done on a regular basis to demonstrate sample integrity and analyte stability after long-term storage (Stephenson et al., 2020)

 In addition, serum metabolite stability depends less so on storage duration and more on the number of freeze/thaw cycles. Past studies have found that most serum amino acids and acylcarnitines remained stable after two freeze/thaw cycle (Breier et al., 2014), which were well within with the scope of current study. Also, all samples were stored and assayed the same way, and analysis was done based on relative comparison of peak signals. Therefore, any relative % changes in molecular quantities would have been consistent and accounted for across the samples. The serum collection/storage information has been updated as follows:

Line 78-79. “Maternal blood samples were collected from non-fasted primigravid participants from the antecubital vein at 28-32 weeks gestation between 2008 and 2011.”

Line 81-83. “Human serum metabolome has been found to remain largely stable at −80 °C within 2 freeze/thaw cycles which was within the scope of the present study.”

References

Stephenson, N. L., Hornaday, K. K., Doktorchik, C. T. A., Lyon, A. W., Tough, S. C., & Slater, D. M. (2020). Quality assessment of RNA in long-term storage: The All Our Families biorepository. PloS One, 15(12). https://doi.org/10.1371/JOURNAL.PONE.0242404

Breier, M., Wahl, S., Prehn, C., Fugmann, M., Ferrari, U., Weise, M., Banning, F., Seissler, J., Grallert, H., Adamski, J., & Lechner, A. (2014). Targeted Metabolomics Identifies Reliable and Stable Metabolites in Human Serum and Plasma Samples. PLOS ONE, 9(2), e89728. https://doi.org/10.1371/JOURNAL.PONE.0089728

  1. Lines 87 and 88:  Although I am sure this data are in your previous publications (13, 14) but for ease of the reader, please state what speed and duration centrifugation was performed here.

RESPONSE: Thank you for the review. The centrifugation speed and duration during sample preparation has been updated as follow:

Lines 97-98: “The mixture was vortexed on a benchtop unit (Fisher brand, USA) set at the maximum level for 1 minute and centrifuged (Eppendorf, Canada. Model 5424R) at 14,000 rpm (18407g) for 10 minutes at 4 °C to condense the precipitate.”

  1. Lines 110-111:  Is there a reference citation for the VIP scoring process?

RESPONSE: Thank you for the feedback. The following reference has been added for the application of VIP scores in a metabolomic study (Cho et al., 2008).

Cho, H. W., Kim, S. B., Jeong, M. K., Park, Y., Miller, N. G., Ziegler, T. R., & Jones, D. P. (2008). Discovery of metabolite features for the modelling and analysis of high-resolution NMR spectra. International Journal of Data Mining and Bioinformatics, 2(2), 176. https://doi.org/10.1504/IJDMB.2008.019097

  1. Line 132:  Should this read < 32 weeks?

RESPONSE: Thank you for the feedback. The statement has been corrected.

  1. Line 152:  I don’t understand this “Error…” statement. Also, see Lines 165-166.

RESPONSE: Thank you for identifying the issue. The messages likely refer to a software error when implementing the citations. This has been updated in the latest version of the manuscript.

  1. Line 278:  Small “t” here.

RESPONSE: Thank you for the feedback. The statement has been corrected.

  1. Line 286:  Should “AUROC” be spelled out here as Tables should stand independent of text?

RESPONSE: Thank you for the insight. The title has now been corrected for the table.

  1. Discussion section:  Can the author(s) add a paragraph commenting/speculating, based on their and the published data cited, if certain supplement strategies (e.g., carnitine; AAs consumption during pregnancy) might be/could be researched and of possible benefit in preventing/minimizing preterm birth.  How might the clinician think about theoretical/possible application of your data in the care of their pregnant patients?

RESPONSE: Thank you for this comment. As alluded to in this manuscript, most metabolite profiles observed in pre-term primigravid women are likely a result of underlying metabolic alterations with the pregnancy/placenta/fetus per se, rather than the mother. Observed changes could be indicative of greater metabolite uptake by the fetus or early placental dysfunction. While proper maternal nutrition is well-known to minimize the risk of pre-term birth and optimize outcomes, suggesting a specific supplement(s) is premature. For this reason, the application/adoption of this work in the health care system is very limited at the present time. We have chosen not to add an additional discussion section on this topic, but rather include it as a limitation and future direction (Lines 361-364) below.

However, the exception to this is acetyl-L-arginine. Our results show lower L-arginine in pre-term birth mothers and there is a rich literature showing supplementation is of benefit. The discussion of this finding, its implications and references have now been expanded (Lines 367-378).

Lines 362-365:  ‘In addition to acylcarnitines, several amino acids and their derivatives were also distinct in preterm birth that may be indicative of greater uptake by the fetus or early placental dysfunction. Future studies are required to decipher the mechanisms behind these alterations and whether preventative measures or dietary supplementation would be of value’.

Lines 367-378. ‘Our finding of lower acetyl-L-arginine in preterm mothers is notable as L-arginine and the nitric oxide system play an essential role in vascular compliance and placental function, on which pregnancy outcomes may be dependent [33]. Dietary arginine assessed by 24-h food recall has been negatively correlated to preterm birth risk with those in the highest consumption quartile having 0.79 times the risk of preterm birth prior to 37 weeks compared to the lowest quartile [34]. L-arginine supplementation also has favorable impacts of birth outcomes, reducing the risk of intrauterine growth retardation and preterm birth [35,36]. As a result, supplementation has been proposed as a recommendation for women at risk of pre-term birth, poor pregnancy outcomes, pre-eclampsia, and other hypertensive disorders of pregnancy [37].  However, in the case of primigravida’s, where information regarding preterm birth risk may be limited, defining, and testing normative acetyl-L-arginine ranges may be of utility [38]’.